# Aspects of Expansive Learning in the Context of Healthy Ageing—A Formative Intervention between Dental Care and Municipal Healthcare

**DOI:** 10.3390/ijerph19031089

**Published:** 2022-01-19

**Authors:** Jessica Persson, Ann Svensson, Ingela Grönbeck Lindén, Sven Kylén, Catharina Hägglin

**Affiliations:** 1Department of Health Sciences, University West, 461 86 Trollhättan, Sweden; 2Centre for Gerodontology, Public Dental Service, Region Västra Götaland, 402 33 Gothenburg, Sweden; ingela.gronbeck.linden@gu.se (I.G.L.); catharina.hagglin@gu.se (C.H.); 3School of Business, Economics and IT, University West, 461 86 Trollhättan, Sweden; ann.svensson@hv.se; 4Department of Behavioral and Community Dentistry, Institute of Odontology, Sahlgrenska Academy, University of Gothenburg, 405 30 Gothenburg, Sweden; 5R&D Department, Primary Health Care, Regionhälsan, Region Västra Götaland, 462 35 Vänersborg, Sweden; sven.kylen@vgregion.se

**Keywords:** oral health, older adults, oral care, work-integrated learning, expansive learning, interprofessional collaboration, interorganizational collaboration, intervention

## Abstract

There are great risks of diseases in the ageing population, and oral diseases are no exception. Poor oral health has profound negative impacts on the quality of life. It is therefore crucial to include the oral health perspective in the care for older adults. To meet the challenges associated with oral health in the ageing population, a formative intervention was launched. The intervention, called the TAIK project (=“Dental hygienist in a municipality organization”, in Swedish: Tandhygienist i kommunal verksamhet), meant that six dental hygienists served non-clinically as oral health consultants in five Swedish municipal organizations. The intervention formed an infrastructure and platform for work that benefits the ageing population and created a new basis for decisions regarding oral health in homecare. The aim of this paper is to explore how aspects of collaboration in an interprofessional and interorganizational intervention may lead to expansive learning. Expansive learning forms the theoretical framework of this study. The dental hygienists and the local head nurses were interviewed individually in-depth. Reflection documents from the dental hygienists were also part of the analyzed data. The conclusion is that the formative intervention was reliant of change which created a foundation for reciprocal understanding that led to expansive learning between dental care and municipal healthcare, with resilience and empowerment as crucial factors.

## 1. Introduction

The average life expectancy is increasing worldwide [1]. The population of Sweden is no exception [2]. The rapidly growing ageing population is a generation with considerably better oral health and more teeth preserved than any previous generation [3]. However, while the ageing population’s oral health is most often good, frailty greatly increases the risk for oral diseases [4,5]. Poor oral health affects the quality of life in many ways, such as difficulties with chewing, eating, and suffering from dry mouth, bad breath (halitosis), pain and infections [6,7,8]. Oral health and general health have a bilateral relationship. Studies show that poor oral health affects the general health in terms of higher prevalence of pneumonia, and increased severeness of already existing diabetes mellitus [9,10]. Connections between poor oral health and cardiovascular diseases [11], involuntary weight loss [12], and malnutrition [13] have also been of interest in recent research. However, the ability to perform daily oral self-care often decreases with age [14]. Ageing individuals can have complex prosthetic constructions that previous generations did not have. Thus, managing oral self-care can be a complex activity. What is also noteworthy is that ageing individuals often lose previous regular dental care contact [15].

Studies have shown that nursing staff often regard oral care work within nursing as unpleasant and difficult [16,17,18,19]. Lack of knowledge and competence regarding oral care within nursing has been expressed. Difficulties with cooperation when performing oral care within nursing, as well as lack of updated and clear routines and regulations, are other important factors that contribute to the difficulties.

The Sustainable Development Goal 3 of the 2030 Agenda [20] describes that it is more and more obvious that there is a need for support systems in healthcare for the increasingly growing ageing population with complex needs of care. In Sweden, however, dental care and healthcare have been described as parallel systems with different management, record systems, education, culture, and economy [21]. To our knowledge, few previous studies have considered the question of oral care in the context of healthcare for older adults from a perspective of learning, strategy forming for healthy ageing, healthcare, and dental care taken together.

In order to establish preventive strategies and support evidence-based decisions regarding oral health for the ageing population in municipal organizations, a formative intervention took place in five Swedish municipalities. The aim of this paper is to explore how aspects of collaboration in an interprofessional and interorganizational intervention may lead to expansive learning. The remainder of this paper is structured as described in the following. First, the theoretical framework is presented. Second, the practical intervention is described, together with materials and methods. Third, the results of the studied intervention are described. Finally, we discuss the theoretical and practical implications of this research, and the paper is then concluded.

## 2. Theoretical Framework; Expansive Learning

Cooperation, meetings, relationships, and understanding between different professionals are of increasing importance in work contexts of today. This implies learning across boundaries and agencies, in an ongoing process of transformation and change of culture in practice at workplaces. Practices consist of a set of interrelated processes of knowledge production, and learning as a recontextualization process of new ideas, experiences, and procedures emerging from the practices [22]. Practice-based theories claim that learning cannot solely be an individual process, as learning emerges from relationships and interactions between people in particular contexts [23]. Here, learning is defined as a process of change or transformation that includes the expansion of possibilities and actions for individuals and groups, and that is related both to knowledge creation and skills development [24]. The work requires horizontal movement and hybridization of learning between different professional domains. This involves collective formation of new concepts, and this type of learning can be called *expansive learning* [25].

Learning is constructed by different practitioners in their collective activity, and the creation of new knowledge goes beyond the perspective of knowledge integration [26]. Individuals actively influence each other’s knowledge and norms through co-participation processes [27]. The content that shall be learned is not well known by the practitioners, who have to cooperate in order to design and conduct new activities. Thus, the design of the new activities and the knowledge and skills it requires are increasingly intertwined [28]. Specific concepts, actions and processes no longer belong solely to certain professions, as they need to be mastered by the breadth of professions involved in the work. In this study, the main focus is interprofessional and interorganizational expansive learning within dental and municipality healthcare. The changes in work require generalizations and learning that expand the learners’ horizons in their practical work [25].

Formative interventions that can result in expansive learning require stimulations, or what Engeström and Sannino call double stimulation. Double stimulation includes a demanding task as a first stimulus and a neutral external artifact as a second stimulus. An individual can fill both tasks with meaning in order to enhance the actions that lead to reframing the task to be performed at work, as expansive learning occurs. In expansive learning, there is a qualitative transformation of the activity system, which includes changes in the behavior and cognition of the learners as practitioners. This is understood as a positive development and improvement in the work performance. The expansive learning theory is based on a notion of double bind, as a social “*essential dilemma which cannot be resolved through separate individual actions alone*”, but where joint cooperative actions can push new emergent activities [25] (p. 5).

Learning processes occur when the subject of learning transforms from individuals to collectives and networks. The object in the activity refers to the problem space at which the activity is directed. Hence, different practitioners share the same general object. However, different professions can have different explicit and implicit regulations, norms, conventions, and standards, which may affect their actions in the activity. As such, the different professions are interconnected in an activity system, with a partially shared object, and constitute a partnership and a network, in which cooperation is ongoing. As the object is shared, the practitioners have to act together to produce their services, even if there is some tension between them. From this situation, a new expanded object and pattern of activity is formed, based on expansive learning. Expansive learning consists of (1) an expanded pattern of activity, (2) a new type of agency, and (3) corresponding theoretical concepts.

## 3. The Intervention

The issue addressed in this paper is the growing ageing population with relatively good oral health (and many preserved teeth), but who, due to increasing frailty, are at high risk of developing oral diseases, making it necessary for dental care and healthcare to find ways to collaborate. The intervention consisted of municipal dental hygienists (MDHs) working part-time (50%) as non-clinical oral health consultants in healthcare organizations in five Swedish municipalities in the Region Västra Götaland, Sweden. The research project lasted for three years and was called the TAIK project. The aim was to ensure and enable healthy ageing with an included oral health perspective. The intervention formed an infrastructure and a platform for interprofessional and interorganizational work that supports healthy ageing.

### 3.1. The Intervention Context

In order to ensure the safety for patients in healthcare, each Swedish municipality employs at least one local head nurse, a MAS (in Swedish; Medicinskt Ansvarig Sjuksköterska) [29]. They serve as local authority senior advisors for healthcare in the municipalities. A MAS has laws, rules, routines and organization knowledge, and also an overall responsibility role for care and nursing activities in the municipality. For example, a MAS establishes routines that enable decision making concerning healthcare in everyday nursing care within homecare.

In Sweden, a dental hygienist is a licensed profession that mainly focuses on oral health [30]. The profession focuses mostly on health promotion and disease prevention [31]. Dental hygienists do not work in the municipal organizations, but in private dental practices, or in public dental care run by the Swedish regions and not by the municipal organizations. Table 1 describes the demographics of the municipalities in which the intervention took place, and the number of local head nurses, MASs, and MDHs in the project [32].

### 3.2. The Core of the Intervention

The intervention formed an interorganizational and interprofessional infrastructure and platform for collaboration that benefits the ageing population of the five municipalities in the project. In order to establish this platform of collaboration, monthly project meetings were held with the MDH, project manager and 1–3 key people from each municipality (for example, the MAS). The key people were chosen by the municipal organizations and participated actively and continually in the intervention (for example, by attending the monthly project meetings in the municipalities). The key people were healthcare professionals with an understanding of municipal healthcare organizations. All municipalities recommended a MAS as a key person in the project, independently from each other. To participate in the project, dental hygienistsapplied for part-time employment as MDHs in the TAIK project. The employments within the intervention were non-clinical, meaning that the MDHs worked as consultants in offices at the municipality organization (together with the key people from the municipality). All MDHs had experience of working with issues regarding oral health for older adults and understood the dental care organizations (both public and private). Monthly project meetings were held in in each municipality and included the key people from the municipality. During the project meetings, the practitioners made plans together based on the defined needs of the ageing population, the municipal organization of the municipality, and the organization of dental care (public or private). Between 2018–2021, a total of 88 interprofessional and interorganizational project meetings were held within the intervention. Monthly team meetings (“team day”) with the project manager and all MDHs were held to share experiences and ensure the quality of the work in order to develop the intervention and support healthy ageing. In total, over the intervention period of three years, 32 full team days were held with the MDHs. A topic could, for example, be: “How do we support the nursing staff in evidence-based decisions regarding oral health in homecare?”

Examples of actions within the intervention (planned interprofessionally and interorganizationally at project meetings and team meetings):Municipal practitioners and MDHs revised routines and checklists from an oral health perspective, in order to help nursing staff make evidence-based decisions in everyday care for older patients in healthcare.MDHs performed auscultations with different healthcare workers from the municipality.MDHs met members of the ageing population at local senior cafés and other events.MDHs and MASs planned lectures after identifying lack of knowledge regarding oral health. For instance, if nursing staff needed to gain knowledge about oral care within nursing care, they defined how they should gain it. Lectures were planned together, interprofessionally and interorganizationally. After lectures were held, they were evaluated together within the intervention. MDHs gave lectures about oral health and practical instructions regarding how to perform oral care to nursing staff in homecare and nursing homes (a total of approximately 2000 nurses and nursing assistants received theoretical and practical education in oral care during the intervention period of three-years).MDHs held workshops with nursing staff to gain knowledge and reflect on their everyday conditions, in order to assure that the population remained healthy while ageing.

Digital activities were held in the last 12 months of the intervention due to the COVID-19 pandemic. During this time, educational films regarding oral care were developed and distributed within the project to the municipal healthcare organizations. The content of the films was defined interprofessionally and interorganizationally within the intervention.

## 4. Materials and Methods

### 4.1. Study Design

This study was conducted as an intervention, by using a qualitative approach. The design is exploratory and has an inductive approach [33]. The research was influenced by the interpretive perspective as it focuses on understanding complexity in the context and in the process of an intervention aimed at implementing a new work practice.

### 4.2. Data Collection

In order to explore aspects of how collaboration in an interprofessional and interorganizational intervention can lead to expansive learning, individual interviews were conducted, and reflection documents were collected and analyzed. The informants were asked to participate strategically, based on their role in the TAIK intervention. They participated in project meetings and initiated the activities described in Section 3.2. The participants shared a common experience as key people in the intervention, but had different perspectives, experiences, and preconceptions (dental care and municipal healthcare). Table 2 describes the selection of informants, and shows that they were all rather experienced. In total, data were collected by individual interviews with six MDHs and five MASs. Written reflection documents from the six MDHs were also collected.

The interviews were performed in a semi-structured way using an interview guide developed to match the aim of the study. The opening question was: “Could you please describe something that you have learned from the TAIK project?” The following questions were also open-ended and related to experiences and collaboration in the TAIK intervention. The interviews lasted approximately 45 minutes and took place digitally, using Skype, at the municipality workplace of the MDHs and the MASs, during the winter 2020–2021. All interviews were recorded and transcribed verbatim.

The MDHs wrote reflection documents in order to describe and reflect on their personal work-integrated learning. They were asked to reflect upon six questions, of which none were mandatory:What have I learnt from the project?How did I learn this?What are prerequisites for the TAIK organization?What would I like to change, if I could do it again?What should be avoided?What had good effects?

The five documents contained between 500 and 5,296 words (Md 997) and were written halfway through the intervention, in May–August 2020.

### 4.3. Data Analysis

In this qualitative study, the data were analyzed using qualitative content analysis and the approach was inductive [34,35]. All transcriptions were made by the same person. In order to systemize the process of analysis, each author individually read through all data in order to get a sense of the whole. Further steps in the analysis were then performed in a constant dialogue with the research group. Data were divided into meaning units and condensed, and the condensed meaning units were coded. Identified codes with similar content were sorted into subcategories. Subcategories with similar content were interpreted and abstracted into categories. Throughout the entire process, the authors discussed the analysis and compared it to the original data, which led to adjustments, before the authors agreed on the final results. Table 3 presents an example from the analysis; the summary regarding the category “Change”.

## 5. Results

Four main categories were identified: “Oral health expertise”, “Understanding”, “Collaboration” and “Change”. Specifically chosen quotes, representative for each category, illustrate the results. Table 4 presents the categories and the subcategories.

### 5.1. Oral Health Expertise

All informants agreed that the availability of a dental hygienist contributed to improving the oral health perspective in the overall healthcare processes in the municipalities. While oral health is considered as a part of general health, in times of priorities and without an available expert, there is a risk that the oral health perspective is neglected or forgotten about. There are great needs for evidence-based information in order to support decision making in the work practice of healthcare in the municipal organizations. Evidence-based knowledge is decomposed into routines and checklists by the MASs, and within the TAIK project, also by the MDHs. To have access to someone with experience and knowledge of oral health was described as an important part of the intervention. Four subcategories were identified: “Responsibility”, “Quality”, “Continuity” and “Availability”.

#### 5.1.1. Responsibility

One key factor regarding the ageing population with complex needs of care in the organization of a municipality, raised by the informants, is the need for clear responsibilities with regard to different assignments. With many different caregivers and perspectives, but no oral health perspective, the dimension of oral health diminishes or disappears, and so does the responsibility. The informants frequently reflected about responsibility:

MAS C: *“Oral health is an important part of the care that the municipal caregivers provide. It should be included—but somehow it is lost. […] The TAIK project has put the mouth on the map”*.

The MDHs expressed surprise at the relatively few routines and strategies present in the municipal healthcare organizations regarding oral health. Despite the fact that the healthcare practitioners considered oral health as important, the oral health perspective was described as being in some ways invisible, as it was not included in general routines and decision support regarding the ageing peoples’ health. The responsibility for oral health in nursing care was regarded as belonging to “everyone and no one”. One informant expressed the feeling that oral health was deprioritized in municipal healthcare. Through the intervention, the informants believed that this dimension of health could become a responsibility for both municipal healthcare and oral healthcare, in collaboration.

#### 5.1.2. Quality

The intervention was regarded to highly affect the quality of healthcare given by the municipal organizations. The absence of information and decision-making processes related to oral health was considered to complicate and delay work, which resulted in a lack of quality in the care offered to the ageing population. The municipality healthcare practitioners considered that the intervention had strengthened their competence regarding assessment of oral health. For example, the MDHs had hands-on education in oral care for nursing staff where the nursing staff practiced how to perform oral care on someone else. In this way, the informants stated that the ageing population received safer healthcare than before the intervention. Higher quality of healthcare was described several times in the interviews.

MAS C: *“We have provided the nurses with mouth mirrors and flashlights to enable them to make oral health assessments. […] That has increased the quality of the oral healthcare given to very many service users”*.

The informants considered it natural that an MDH should be part of the municipal healthcare organization, since most municipal healthcare practitioners perceive oral health and healthcare as difficult, but important.

#### 5.1.3. Continuity

A key factor for the intervention was the regular meetings with practitioners from the municipality and dental care. It was described as something rather unique, and as a dynamic process that required “Continuity”.

MDH B: *“My attitude has changed from that I should, kind of, try to change something to that, we should do this more together* [dental care and municipal healthcare]. *[…] I probably hadn’t really realized what a challenge it is that we live longer, we will get older, and if dental care makes an effortwe need to do it together. […] If you are on site, you have time, because I don’t think there is any simple solution that will just happen. We need to work with it* [oral health in municipal healthcare] *all the time”*.

Feeling included in a community and having access to a network with competencies were regarded as prerequisites for continuing the collaboration. Having opportunities to talk to other people who work in the very same context, and sharing knowledge between practitioners, were thought to support the “Continuity”. It was stated that in order to develop sustainable change, the practitioners need to work together and respect each other’s work domains.

#### 5.1.4. Availability

The inclusion of MDHs in the municipal organizations on a daily basis was considered to make the oral health perspective more available and visible. For the MDHs, the interprofessional processes of care also became more available and easier to work with. Practitioners representing different perspectives were available in the everyday work life, and they were easy to get in touch with, in a natural way.

MAS A: *“When they are in the building, you know where to go and ask. They are available and they eat lunch here. It makes it more natural in some way”*.

The MDHs expressed satisfaction of being included in the daily work of caring for the older adult, and of being available and having opportunities to influence the development of materials used for making decisions, which also considered oral health.

### 5.2. Understanding

The practitioners in the study described how the intervention contributed to a deeper understanding that could be used as support when making decisions in the daily work with the older adults. Better possibilities to understand different perspectives and each other’s different organizations contributed to making the intervention dynamic. The different conditions in combination with increased understanding of how different practitioners were working made the intervention grow. This understanding made it considerably easier to address messages in a correct way, to the correct person, or to the correct process. Defined subcategories were “Customized messages”, “Shared perspectives” and “Knowledge of ageing”.

#### 5.2.1. Customized Messages

The informants described how they “learned from the learner” within the project. They understood the views of different people and adapted their messages to different receivers, instead of repeating their point of view in the same manner all the time. “Customized messages” led to the development of new activities.

MDH B: *“If you are going to talk to an association for retired people, there are so many different ways to reach them. Everything from making a PowerPoint presentation or a quiz walk, to sitting down and having coffee with them while having a dialogue”*.

Different types of information materials, such as routines, checklists, and different digital materials, were created and customized for different groups, for instance the ageing population, temporary and hourly employees, as well as unit managers in municipal healthcare. In order to support the care for the ageing population, the MDHs meant that the oral health perspective needed to be adjusted to the context in which it was going to be applied. Information and materials that are used to make decisions need to be updated and adjusted, especially with regard to whom is responsible in the complex healthcare processes. For instance, many informants expressed a great desire to support decisions regarding oral health issues, for example by defining whom to contact and when, regarding oral health symptoms, in order to improve the healthcare quality.

#### 5.2.2. Shared Perspectives

The intervention led to sharing of perspectives in new ways. Even though the informants agreed that oral health is a part of the general health, and also a part of the daily general care, the informants stated that they had gained an understanding and shared perspectives in ways that they had not been able to do before. It was challenging for the MDHs to understand the division of work and the work processes in the municipalities. The MASs were challenged especially by the importance of the oral perspective in their own work. For example, the medical record systems used in dentistry and municipal healthcare are different, and cannot be reached from the other organizations. Furthermore, oral health tended to be invisible in the pre-defined content of the implementation plans for the older adult receiving care from the municipal organizations, or at the same care level as brushing hair. The MASs as well as the oral MDHs learned something from each other that they did not know before.

MAS E: *“But it also took, initially at these meetings, it took an incredible amount of time to understand each other. […] What can we do for each other?”*

In order to formulate new strategies and make oral healthcare more present in the decision-making processes concerning healthy ageing, the informants expressed how they needed to “Share perspectives”.

#### 5.2.3. Knowledge of Ageing

Mainly the MDHs described an overall understanding and increased “Knowledge of ageing” in general. The intervention led to meetings with members of the ageing population. These meetings were always professional but non-clinical, which was described as a new experience. The meetings with members of the ageing population, in combination with interprofessional meetings in the municipal healthcare organizations regarding the ageing population, led to a deeper level of understanding of what ageing really means.

MDH A1: *“I have gained an increased understanding of ageing. Getting older and being taken care of. Before, I have maybe only seen it from the outside or how should I say, as a relative or so. Now I see it with a deeper understanding”*.

The MDHs stated that this “Knowledge of ageing” helped them form strategies within the TAIK project. Taken together, the increased “Knowledge of ageing” supported the MDHs in discussions regarding what decision support nursing staff working in homecare need with regard to oral health and healthcare.

### 5.3. Collaboration

The ageing population of 65+ is a heterogenous group. While ageing, many older people enter different “arenas” such as pharmacies, homecare, social events, and different networks. The informants described how they used the TAIK intervention as an infrastructure and a platform for interprofessional and interorganizational work, as it gathered representatives from many different arenas and functions of the society. Together they formulated actions and overall common strategies in order to achieve a common goal: healthy ageing that considers the dimension of oral health. Four subcategories representing different aspects of collaboration were defined: “Dialogue”, “Personal relations”, “Network”, and “Common goals”.

#### 5.3.1. Dialogue

Integrating a new function within a municipal organization was a task that many informants described as challenging. They expressed that it was necessary to have dialogues in order to share knowledge, by, for instance, observing, listening and also speaking. These continuous dialogues were considered crucial for the growth of the intervention, and for proceeding with the activities in the project.

MDH C: *“You are part of a context and you get a sense of, oh no, now we need more! It’s possible to have a dialogue”*.

Different kinds of dialogues took place. Healthcare practitioners from the municipalities were informed about the municipal processes in healthcare at team meetings, which also included common time for questions and reasoning. In the same manner, the MDHs provided education and workshops for the municipal healthcare practitioners, with time for questions and common reflection.

People with many different professions are working with ageing populations and are involved in making decisions regarding their healthcare. Care administrators, homecare practitioners, and relatives sometimes need help to make decisions, also based on knowledge about oral health perspectives.

The MDHs expressed that, when something is to be implemented, it has to be processed at different levels of the municipal healthcare organization. This means that it could take a relatively long time to process specific matters and reach decisions, as the matters have to go through all stakeholder levels. An ongoing “Dialogue” with the different stakeholders was considered to facilitate this process.

#### 5.3.2. Personal Relationships

Actions that led to activities in the intervention were formulated in collaboration with specific key people. As stated before, the process of ageing can involve many different arenas, and the MDHs described that the people to contact and work with sometimes changed. The informants emphasized the importance of “Personal relationships” in the project.

MDH A1: *“But to just sit in the lunch room and establish a relationship before the meeting, that has been exciting. Later when we met at the meeting, we have dared to meet a little bit outside, before”*.

For example, suggestions that are not thoroughly considered by all stakeholders are less likely to be implemented. However, working together with several other competencies provided opportunities for tackling problematic situations with a broader perspective. Good contacts and relationships among the different practitioners facilitated collaboration.

MDH A2: *“We had quite a lot of work with revising the routines. That meant that functions from other parts of the municipal administration came to the administration of the healthcare for older adults. So other administrations participated, because the routines actually concern everyone. We could connect with other parts of the organization that were not in contact with the dental care”*.

#### 5.3.3. Network

To form and establish networks was an important part of the intervention. The informants meant that it was necessary to establish networks in order to work interprofessionally, and networks were created, both within the municipal organizations and within dental care.

MDH B: *“We have CÄT [Centre for Gerodontology] and the Swedish Public Dental Care to lean against if you have questions. Questions can be passed on, so that it is not only towards the municipality, but you also need support from the Swedish Public Dental Care”*.

When discussing aspects of a various issues, it is important to have a network and key professions who support suggestions regarding how practices can change. The MDHs considered it important to have access to different professions in line when considering if a new suggestion could be a sustainable solution.

The power of having access to the MDH network was also considered important.

MDH A2: *“That we have been able to be sounding boards for each other and that we have been able to share. If a MDH has managed to implement a routine, then we have perhaps been able to inspire and support each other”*.

#### 5.3.4. Common Goals

Many informants described how activities in the intervention aimed to achieve “Common goals”. These goals were formulated together, after defining and sharing common problems. These common goals were also evaluated and discussed together, which led to actions and activities that were formulated and planned together.

MDH D: *“The municipality and TAIK formulated three goals. We should increase the number of N-certificates (an economic support/aid concerning dental care for individuals with major needs of care in daily life), increase the knowledge about oral health among healthcare professionals and nursing staff, and increase the number of ROAG (ROAG–Revised Oral Assessment Guide, an oral health assessment instrument for use by healthcare and nursing practitioners [36] and assessments in Senior Alert (a preventive Swedish quality register for people aged 65+ [37])”*.

The MDHs described how the municipal healthcare managers asked for more information regarding oral healthcare for healthcare substitutes in homecare. Assistant nurse supervision programs in upper secondary school formulated common goals together with the MDHs concerning basic oral care. Municipal healthcare practitioners and MDSs developed new flowcharts about care processes. All in all, these actions supported decisions regarding oral health in everyday care for the ageing population. One MDH expressed that these actions also increased the healthcare practitioners’ confidence to make decisions and do their work.

### 5.4. Change 

The informants described different dimensions of “Change”. The work of integrating a new profession was a change in many dimensions, both for the organization, as well as for the key people who participated in the intervention. Everyone faced new questions that challenged them and their organizations. This led to work-integrated learning in many ways, and also to changes in the roles of both the dental hygienists and the municipalities, with regard to oral health in nursing care.

#### 5.4.1. Learning

To share experiences and reflect together from different points of view led to increased learning in everyday life. Including a new perspective into the organization, and working together for several years, set the ground for work-integrated learning. New activities could be developed together based on this learning. The different professions learned from each other. During the pandemic, for example, the intervention supported healthcare practitioners with knowledge about the importance of oral care when the older adults had problems with swallowing. When the older adults could not come to dental clinics during the pandemic, the MDHs made a checklist regarding basic oral care in nursing. The aim of the document was to support decisions regarding daily oral care. The MASs distributed the checklist to all healthcare professionals in the municipalities. The foundation for learning that the intervention created made it simple to support each other.

The informants expressed how integration of oral care in healthcare became even more important than before.

MDH B: *“It may not always be so easy for the municipality either—where does the oral health fit? TAIK also contributes in this way, making the oral health visible on various arenas, which they themselves may not have thought about”*.

If anything in the work is new and unclear, there will be reactions. Having access to someone to consult in different situations, who can provide relevant information, makes situations more manageable. Therefore, the informants considered it natural to have a practitioner with competencies in oral health who can contribute to the knowledge in municipal healthcare. The MDHs expressed how learning also occurred during reflections and discussions at meetings about the educations and their effects. Discussing different possibilities and making priorities is associated with ongoing learning.

#### 5.4.2. Progress

Another part of “Change”, as the informants described it, was “Progress”. This sense of progress made the MDHs feel like they were really included and “welcome in the club”. They felt that they were collaborating with the municipal health practitioners, also when they highlighted the oral health perspective. Moreover, routines and different documents had been updated to better include oral health.

MDH A2: *“Now we feel like any organization developer and it is much easier to make ourselves heard compared to before. Before, I felt that “yes, but you are the dental people, now you can take over everything that has to do with the teeth”. But that’s not the point. They should be able to look at the mouth like any other part of the body”*.

The informants stated that it is important to develop routines and guidelines that can support the change. Method developers got involved in the intervention to inform about decision guidelines concerning the ageing population and different municipal perspectives. Coordinators at nursing homes contributed with knowledge on how the work is performed. A dietician was also involved to include another important area of competence. Several key competences were thus engaged to inform healthcare decisions for the ageing population. To find such competences is of vital importance for the motivation and inspiration to continue the change However, the strict hierarchy in the municipal healthcare organization was considered a hindrance.

#### 5.4.3. Innovation

In order to include the perspective of oral health, many informants described that they needed to be innovative, consider new opportunities and in many ways be unafraid and willing to try. The informants were also aware of the importance of knowledge in pedagogy, in order to have tools to explain, mediate, and negotiate messages based on their own competence.

MDH B: *“As a MDH, you have to be a problem solver and inventive. It is good to have such a side, because the answer is not always there. You need to have an ability to handle things and try to solve them”*.

Awareness of mutual knowledge and the importance of different perspectives of knowledge, made the participants in the study realize that collaboration is necessary in the healthcare of the ageing population. Creativity is stimulated when working together. It is, however, important to have common routines and fixed work structures.

## 6. Discussion

### 6.1. Expanded Pattern of Activity

The intervention led to the establishment of arenas for continuous collaboration and reconciliation between dental care and healthcare, for a period of three years. As the intervention progressed, throughout the three years, new actions were developed together in collaboration. The knowledge which led to the actions in the intervention could not have been foreseen from the start, because the practitioners had to learn and understand things, before taking the next step. In the subcategory “Quality”, one MAS described how the municipal organization had purchased mirrors and flashlights in order for nursing staff to be able to examine the oral status. It might be regarded as a simple task, but if such activities are not planned in collaboration, they might easily be overlooked, since they are not “on the agenda”, and since the healthcare practitioners in the municipality might never have thought about how to best examine a mouth.

The informants concluded that it is good to be innovative and have the abilities to create “Personal relationships”, “Share perspectives”, and have “Dialogue”. Altogether, those seem to be important elements in order to expand the pattern of activity. Engeström and Sannino describe how formative interventions can expand into previously unknown contents [25]. Here, the focus is the agency among the participants and the key outcome of the intervention. The process is owned and led by the practitioners. The results indicate that the practitioners are able to cope with a new pattern of activity—a prerequisite for expansive learning. They seem to have gained a new pattern of activity from developing resilience in their work.

### 6.2. New Type of Agency

The implementation of a new profession in an established organization could be regarded as an intervention where two worlds collide. In the middle of everything there are six MDHs and five MASs who regularly meet and share perspectives. One category in the results is “Change”, and even though the common goal was always healthy ageing, the road to healthy ageing was described by the informants as revised and adjusted over and over again in the intervention. Engeström and Sannino state that dilemmas, conflicts, critical conflicts, and double bind are important discursive manifestations in the analysis of formative interventions that aim to create organizational change and expansive learning [38]. An example of expansive learning could be the representative quote from an MDH in the subcategory “Continuity”. She stated that the intervention had made her realize that there are no easy answers to this question, but that we have to do it together. Engeström and Sannino especially highlight the double bind when an individual realizes that she or he cannot manage the change alone, instead it leads to actions, a transition from the individual “I” to “we” [38].

The results of this study indicate that the absence of an oral health expert who understands the municipal organization tends to make the oral health perspective invisible in municipal healthcare. This possibility of interorganizational learning in the intervention, sharing of perspectives, and using the intervention as an infrastructure and platform for healthy ageing, are essential elements that should be considered when planning care for the increasingly growing ageing population with complex needs. The aspect of “Change” cannot be emphasized enough. Two current articles, Botngård et al and Gustavsson et al, describe oral care in nursing homes and in palliative homecare as an invisible dimension and an area with great need for improvement [39,40]. Previous research suggests that the oral health perspective is largely invisible in nursing due to lack of knowledge among nursing staff [41,42]. Others argue that it is the attitude towards oral health among nursing staff that is problematic [16,17,18]. The results of the TAIK intervention highlight that this is a complex issue in which knowledge and attitudes are two very important aspects in a wider context of ageing and the organizational structures that should ensure healthy ageing. By continuously working together and being available for each other, MASs and MDHs identified what constitutes healthy ageing with an included oral health perspective, in a dynamic, ongoing, and collaborative process. As one MDH expressed it: “*I don’t think there is any simple solution that will just happen. We need to work with it all the time*”. The results indicate that this question can be addressed successfully both interprofessionally and interorganizationally.

The results also show that it is possible to establish expansive learning both interprofessionally and interorganizationally, which leads to safer healthcare in homecare with safer routines and checklists regarding oral health. The professionals also felt more confident to act and learn, even when the work tasks were outside of their immediate responsibilities. Thus, expansive learning in such new arenas as this collaborative infrastructure appears to require empowerment for learning, which the practitioners gained.

### 6.3. Methodological Reflections

Qualitative content analysis was used in order to describe the variation regarding the aim of the research [34,35]. We focused on the subjects (the practitioners) and the context in which the activities took place. Graneheim et al describe how the analysis is a co-creation between the data and the researchers [35]. When it comes to dependability, the previous understanding of the research question in the group was a strength. Two researchers had experience of working with the intervention and three had no experience. Two authors are not working in dental care. The authors without a background in dental care were experienced researchers within work-integrated learning, change and expansive learning. They added a depth concerning the theoretical framework and were also highly involved with the work of developing the interview guide and interviewing. This mix of experiences broadened the discussions throughout the work with this paper, both with regard to planning the interviews and analyzing the data.

Different perspectives of expansive learning run through all the categories, in different shapes. It is also the theoretical framework of this paper. Graneheim and Lundman state that human experiences tend to be entwined, and therefore it is not always possible to create mutually exclusive categories. The level of abstraction of the results is a sensitive task. Interpretations that are too abstract or too concrete tend to be meaningless [34]. We have strived to keep the interpretation on a concrete level, maintaining the words of the informants to a high extent. This was mainly done in order to make the results more useful as a basis for further similar interventions.

## 7. Conclusions

This paper aimed to explore how aspects of collaboration in an interprofessional and interorganizational intervention may lead to expansive learning. The conclusion is that the intervention created a foundation for a reciprocal understanding, through change and led to expansive learning between dental care and the municipal healthcare. The intervention studied here involved interprofessional and interorganizational collaboration. The clinical implication of the intervention was more secure healthcare in which the oral health perspective was more visible; a perspective that could be regarded as contributing to a more holistic view of health than before the intervention. The practical implication of the interventions seemed to be that interprofessional and interorganizational expansive learning occurred, through the established collaborative platform. The theoretical contribution of the TAIK intervention is that the practitioners seemed to gain resilience and empowerment in the process of expansive learning, which enabled them to establish a changed arena for more healthy ageing. These factors seem to be crucial prerequisites for expansive learning.

In a world with an increasingly growing ageing population with complex needs of care, more research is needed with regard to the dimensions of health and expansive, work-integrated learning. In order to form long-lasting, dynamic plans in healthcare, and for instance support evidence-based decision making in homecare with a perspective of expansive learning, it is important to include the perspective of change in further interventions and research regarding oral health and healthcare, as well as organizational aspects of the issues.

## Figures and Tables

**Table 1 ijerph-19-01089-t001:** Characteristics of the five municipalities that participated in the intervention: the numbers of inhabitants, percentage of inhabitants 65+, and number of MASs (=Local head nurses) and MDHs (=Municipal dental hygienists).

Municipality	A	B	C	D	E
Inhabitants (*n*)	113,714	47,050	59,249	56,791	24,513
Inhabitants 65+ (%)	19.2	20.1	19.1	19.6	26.3
MAS	3	1	1	1	1
MDH	0.5 + 0.5	0.5	0.5	0.5	0.5

**Table 2 ijerph-19-01089-t002:** The informants, gender, and years of practice.

	MDH (*n* = 6)	MAS (*n* = 5)
Female	100%	100%
Years of practice as dental hygienist/nurse	Md 29 (min 4, max 35)	Md 32 (min 12, max 39)
Years of practice as MDH/MAS	Md 2 (min 2, max 3)	Md 10 (min 1, max 12)

MDH = Municipal dental hygienists, MAS = Local head nurses, Md = median.

**Table 3 ijerph-19-01089-t003:** Summary from the analysis regarding the category “Change”.

Condensed Units of Meaning	Codes	Subcategories	Category
Adequate * level of education and ability to follow-up practically. Organizational learning leads to a snowball effect, which is not possible with a single occasion with information. It is part of the concept to question.	• Workplace learning • Organizational learning • Questioning	Learning	Change
Dare to evaluate. Put it so that it falls in fertile ground. It is the way of the TAIK project, we do not have all the answers.	• Think critically • Be strategic • Have faith in the TAIK intervention	Progress	
You have to be inventive and creative. There are no simple answers. Reflect on new or different ways.	• Be inventive • Be solution oriented • Be creative	Innovation	

* for instance, the ability to perform basic daily oral care on someone else.

**Table 4 ijerph-19-01089-t004:** Identified categories and subcategories.

Category	Oral Health Expertise	Understanding	Collaboration	Change
SUBCATEGORIES	Responsibility	Customized messages	Dialogue	Learning
	Quality	Shared perspectives	Personal relations	Progress
	Continuity	Knowledge about ageing	Network	Innovation
	Availability		Common goals	

## Data Availability

Due to the nature of this research, participants of this study did not agree for their data to be shared publicly, so supporting data is available from the authors with the permission of the informants.

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
