# Peer review of "Aspects of Expansive Learning in the Context of Healthy Ageing—A Formative Intervention between Dental Care and Municipal Healthcare"

_ijerph, 2022, doi:10.3390/ijerph19031089_

Round 1
Reviewer 1 Report
The aim of this qualitative study is to explore aspects of how collaboration in an inter-professional and inter-organizational intervention can lead to expansive learning. The authors conclude that the work of change created a foundation for reciprocal understanding that led to expansive learning between dental care and the municipal healthcare organizations.
My comments/suggestions for the authors are outlined below.
Abstract
Please define the abbrevation: TAIK.
Introduction:
Please give more detailed information on expansive learning you want to investigate, e.g. calibrating nursing staff or patients. And specify the meaning of expansive learning in context of oral health.
Please clarify your interventions, e.g. SOPs for the MDH and MAS. Are the given lectures and workshops standardized in order to compare?
Materials and Methods:
Please describe the methods of analyzing the qualitative data, eg. transcription, systematization.
How many and which interventions were performed by each MDH and MAS? How many nurses were included?
Table 3: Please clarify the definition of adequate level of education.
How did the pandemic situation influenced the project?
Results:
Present your results adapted on the methods you use of analyzing the quantitative data.
Table 2 is not informative; any characteristics were missing or describe the years of practice in the text. Are there statistically diverences?
Discussion:
Line 503: Please mention the period of three years in the Material and Methods.
Please outline the clinical, theoretical, or practical implications of the reported results and/ or the further need for the Swedish health care system, e.g. possible financing options.
Author Response
Thank you very much for reading and commenting on this manuscript!
Please see the attachment.

Reviewer 2 Report
Aspects of expansive learning in the context of healthy ageing – a formative intervention between dental care and municipal healthcare
This study explored aspects of how collaboration in an interprofessional and interorganizational intervention can lead to expansive learning. Although the topic and results are valuable, there are several points required reconsideration in this manuscript.
The overall description is very long, so please be concise where you can concisely describe, especially in the results section.
Section 3.2. and 4.3. are lacking in the manuscript.
L149 Please describe what kind of role the key people are and how they were selected.
L188 The number of years of experience in MAS varies, but is there any effect of this variation?
L189 I recommend that you specify the 11 interviews in a table.
L189 Please describe how you decided on the content of the interview.
L571 In this intervention, authors who do not work in dental care and authors who do work, authors who have experience in intervention and authors who have no experience in intervention jointly analyze and analyze. I am concerned that the opinions of experienced dental professionals will be mainly reflected. Please describe how did you respond to such concerns.
L579 Specifically, how did you strived to keep the interpretation on a concrete level?
Conclusion
The statement of conclusion is too long, including some discussions. In the conclusion session, please be sure to briefly describe the conclusion.
Thank you for giving the opportunity to review this manuscript. I hope that the authors revise and reply to the stated points and review again.
Author Response
Thank you very much for reading and commenting on this manuscript!
Please see the atteachment.

Round 2
Reviewer 1 Report
Please add line 231 before table 2.
Text editing is needed: e.g. line 643.
Layout Table 3: Category: working change belongs to what?